# Effect of Temperature on Microwave Permeability of an Air-Stable Composite Filled with Gadolinium Powder

**DOI:** 10.3390/s22083005

**Published:** 2022-04-14

**Authors:** Sergey N. Starostenko, Dmitriy A. Petrov, Konstantin N. Rozanov, Artem O. Shiryaev, Svetlana F. Lomaeva

**Affiliations:** 1Institute for Theoretical and Applied Electromagnetics, Izhorskaya 13/19, 125412 Moscow, Russia; dpetrov-itae@yandex.ru (D.A.P.); k_rozanov@mail.ru (K.N.R.); artemshiryaev@mail.ru (A.O.S.); 2Physical-Technical Institute, Udmurt Federal Research Center UB RAS, T. Baramzina Str., 34, 426067 Izhevsk, Russia; lomayevasf@mail.ru

**Keywords:** microwave permeability, Curie temperature, cluster magnetization, Hopkinson effect, mixing model, tunable screen

## Abstract

A composite containing about 30% volume of micrometer-size powder of gadolinium in paraffin wax is synthesized mechanochemically. The composite permittivity and permeability are measured within the frequency range from 0.01 to 15 GHz and the temperature range from ~0 °C to 35 °C. The permittivity is constant within the measured ranges. Curie temperature of the composite is close to 15.5 °C, the phase transition is shown to take place within a temperature range about ±10 °C. The effect of temperature deviation from Curie point on reflection and transmission of a composite layer filled with Gd powder is studied experimentally and via simulation. Constitutive parameters of the composite are measured in cooled coaxial lines applying reflection-transmission and open-circuit-short-circuit techniques, and the measured low-frequency permeability is in agreement with the values retrieved from the published magnetization curves. The effect of temperature on permeability spectrum of the composite is described in terms of cluster magnetization model based on the Wiener mixing formula. The model is applied to design a microwave screen with variable attenuation; the reflectivity attenuation of 4.5 mm-thick screen increases from about −2 dB to −20 dB at 3.5 GHz if the temperature decreases from 25 °C to 5 °C.

## 1. Introduction

Adaptive composites with tunable constitutive parameters look promising for many applications including microwave screens, sensors, waveguide elements, etc. [1,2]; however, few of them are used in practice. One of the problems is that the tuning needs electric bias about 10–30 kV/cm for ferroelectrics [3] or magnetic bias about 2–4 kOe for composites with permeable microwires [3]. Here the temperature tuning of microwave permeability *µ* is under consideration. It may seem that such tuning is possible at a temperature of several hundred degrees centigrade and has too high a transit time compared to bias tuning. The temperature may be decreased by the proper selection of a magnetic substance; the time for heating or cooling for several degrees is comparable to the transit time for application of several kOe bias.

Gadolinium (Gd) is a ferromagnet with a Curie temperature of about 20.2 °C, and its saturation magnetization is 15% higher than that of iron, therefore powders of Gd and its alloys look like promising fillers [4,5] for microwave applications, especially as adaptive composites with parameters governed by temperature.

The magnetization of bulk gadolinium has been thoroughly studied, as Gd is the first material used for magnetic refrigeration; nevertheless, the published magnetization curves [6,7,8] differ significantly and there are practically no data on gadolinium permeability. To retrieve at least a reference value of permeability *µ* to compare with the measured data a set of published magnetization curves is digitized and the static permeability is estimated as:(1)$$\mu_{static}\underset{H\to 0}{\approx }1+M/\left[\left(1-N\right)\times H\right]$$

Here *M* is the sample magnetization for the strength of magnetic bias *H*, *N* is the demagnetization factor (the shape factor) defined by sample shape, and orientation is relative to bias direction [6,7]. The shape factor range is 0 ≤ *N* ≤ 1, for spheres *N* = 1/3 [9,10].

The magnetization data at low bias are scarce, but the hysteresis loop looks narrow, so the estimations with Equation (1) should be valid up to a bias of about 18,000 G [11]. The problem is that the higher the bias, the higher the observed Curie temperature (see curve d in Figure 1). Another problem is that monocrystalline gadolinium is anisotropic; the magnetization curve may depend on the sample shape and orientation [6,12]. The retrieved susceptibility χ=μ−1 curve (Figure 1, line a) is calculated as a mean value for two orientations of easy magnetization axis. The susceptibility estimates with Equation (1) (Figure 1) show that the permeability of polycrystalline Gd is much lower than that of iron-group metals. Below −100 °C μ≈4÷7, at *t* ≈ 0 °C the permeability is about μ≈1.5÷4 and falls rapidly with temperature increase (Figure 1).

The published data on high-frequency permeability of gadolinium were obtained in a wide temperature range with two techniques: the low-frequency impedance measurements of a bulk rod of gadolinium [13,14,15] and the ferromagnetic resonance (FMR) measurements of tiny Gd spheres at 9 GHz [16,17] or monocrystalls at 32 GHz [18]. The impedance measurements [13,14,15] performed in the frequency range below 100 kHz are very sensitive to the purity and structure of the metal surface, therefore the obtained results depend on sample treatment and differ significantly for various samples. Moreover, the calculations [13,14] neglect a slight conductivity change of Gd in the vicinity of Curie temperature [19] and result in permeability values (µ ≈ 1400 + i300 at 10 kHz and −63 °C [13]) about 1000 times higher than the estimates from magnetization curves (Figure 1).

The samples for FMR measurements are usually paraffin-bound composites filled below 1% vol of Gd [16,17]. The measurements result in the relative dependence of permeability on external bias; recalculation of the field-domain data into the permeability dependence on frequency using LLG formula [20] leads to rough estimations only. 

The problem of permeability dependence on temperature for composites that are filled with particles of various shapes and sizes is that the Curie temperature of the particles is not a constant equal to 20.2 ÷ 21 °C [21], but there is a dispersion of the measured temperatures depending on many factors that are difficult to define (inhomogeneity of atomic structure [22], the random orientation of easy axis and domains in a polydomain sample [17], etc.). The most essential factors are accounted for by Equation (2), derived from analysis of the Brillouin function [23]:(2)$$T_{C\_meas}\left(N\right)\underset{H_{ext}\neq 0}{=}T_{C}\left(1-\frac{J+1}{3J}\frac{\mu_{B}\left(M_{0}N-H_{ext}\right)}{kT_{C}}\right)$$

Here *T_C_* is Curie temperature (°K), *H_ext_* is the strength of external bias, *J* is the spin magnetic moment, *µ_B_* is the effective Bohr magneton number, *N* is demagnetization factor (shape factor) defined by inclusion shape, *k* = 1.380649 × 10^−23^ J/K is the Boltzmann constant, *M*_0_ is the saturation magnetization.

For zero bias (*H_ext_* = 0) Equation (2) shows that the measured Curie temperature linearly decreases with the increase of demagnetization factor *N* and cannot exceed 293.2 °K (20.2 °C). The minimal Curie temperature is obtained for flakes or films perpendicular to the external field (with *N* ≈ 1); the maximal temperature is obtained for fibres or films parallel to the external field (with *N* ≈ 0). The maximal difference of the measured Curie temperature from 20.2 °C due to particle shape for *J* = 7/2, *μ_B_* = 7.55 and *M*_0Gd_ = 2.42⋅× 10^6^ A/m is about 5°. 

The values of *T_C_*, *M*_0_, *µ_Bohr_* and even *J* for Gd in Equation (1) are not known as accurately as Zverev [23] has taken for the above estimate, e.g., the effective Bohr magneton number in for Gd varies according to published data from 7.55 to 7.98 [19,24]. The saturation magnetization *M*_0_ is a distributed value because of spin-waves, defects of crystalline structure, random orientation easy axis and domain magnetization [19,21,22]. As result the reported Curie temperature falls out of the 5 °K range defined in [23] (e.g., the measured Curie temperature for Gd films is about 44 °C [25]) and the dependence of low-frequency permeability of Gd on temperature is neither a step-like drop, nor a linear decrease according to Equation (1) to *χ*_0_ = 0 at *T = T_C_*, but looks like a flattened distribution curve with the maximal density close to *T_C_* values. The similarity is confirmed by the dependence of coil inductance as a function of temperature, where the coil core is a dispersion of Gd-filings in oil [26]. 

Neither the static permeability of Gd, nor the permeability dependence on frequency have been studied. Therefore, the research aims are to synthesize practicable amounts of air stable composite with Gd powder (the synthesis a challenge itself as the fine powder is pyroforic in air); to study the dependence of constitutive parameters of the composite on frequency and temperature (the effect of temperature is here an additional factor to refine mixing rules) and to estimate performance of the composite as a microwave screen with temperature-tunable attenuation. 

## 2. Synthesis of Gd Powder and Preparation of Composite Samples

Commercial Gd of 99.6% purity is available in shape of 3–4 kg ingots. Synthesis of a composite filled with micrometer-size powder of Gd is a chemical challenge itself, as Gd powders are pyrophoric in air. The developed technique, in contrast to the vacuum vaporisation technique [16,17], is aimed to produce amounts of micrometer-size gadolinium powder sufficient for preparation of samples for free-space microwave measurements (up to about 50 g of powder per cycle). Similar to the published procedure [27], the technique is based on the mechanochemical treatment of bulk gadolinium.

The ingot is mechanically lathed into shavings of about 0.5 mm thickness. The shavings are grinded for an hour in a Fritsch P7 planetary mill with 12 mm steel balls in a steel mortar, and KCl is added into the mortar as an abrasive agent. To prevent ignition of Gd particles, the grinding is performed in argon media. The grinding product is washed with an alcohol-water mixture to dissolve KCl, then dried with acetone and stored in petroleum-ether to prevent oxidation. The synthesized powder slowly oxidizes in air, but readily self-ignites on kicking or rubbing. 

The powder microstructure, morphology and impurity distribution is analyzed with a scanning electronic microscope (SEM) VEGA 3 LMN (TESCAN) equipped with an energy-dispersive X-ray spectroscopy (EDS) system. SEM images are shown in Figure 2; EDS shows traces of surface oxidation and carbidisation of obtained powder. The images show that the powder consists of particles of various shapes and sizes, the small particles have an approximate uniform size of about 0.1 mcm along three axes, while the larger particles are close to flakes of about 2 mcm diameter. The flakes are partially agglomerated into clusters up to 20 mcm in size.

X-ray phase analysis of the obtained powder is performed with a MiniFlex (Rigaku) X-ray diffractometer using Co Kα radiation. An X-ray diffraction powder pattern with main Gd and GdH_2_ peaks indexed is shown in Figure 3, and the measured and reference data on peak position, intensity and corresponding Miller indices (hkl) are shown in Table 1. The analysis shows that the obtained powder is metal Gd with about 10% vol. additions of gadolinium hydride GdH_2_. The increase of grinding time leads to the partial transformation of metal into GdH_2_.

The paraffin-bound composite with 30% vol ($p_{incl}=0.3$) of the above powder is prepared by mixing it in an ultrasonic bath heated to 100 °C. The volume fraction (the filling factor) *p_incl_* of Gd in a composite is calculated accounting for a GdH_2_ impurity of 10%. The composite is stable in air (there is neither a weight increase nor constitutive parameter change in the washer-shaped samples stored for more than a year at room conditions).

## 3. Measurement Techniques

The complex microwave permittivity *ε* and permeability *µ* of samples under study are determined with a vector network analyzer (Keysight N5224B) in standard 7 × 3 mm coaxial cells. At the frequency range 0.3–15 GHz, the measurements are performed in a coaxial airline applying the reflection-transmission technique. The thickness of washer-shaped samples is about 1.5–3 mm. 

At the range 10–300 MHz, the measurements are performed in a cell with a detachable short applying short-circuit technique for permeability measurements and open-circuit techniques for permittivity measurements. The sample thickness is about 10–15 mm, and the washers are additionally pressed inside the cell to reduce the air gaps formed while the sample is slid into the cell. 

The above supplementary techniques are applied to improve the accuracy of low-frequency data and to decrease the effect of air gaps on the measured permittivity. The static *ε* and *µ* are determined as median values in the frequency range of 10–200 MHz where the loss is negligible.

The transmission cell is equipped with an external ring-shaped cooling jacket; the cooling is performed by a flow of nitrogen vaporized from a Dewar vessel. The obtained maximal cooling is about −190 °C; the heating above the room temperature is performed with a flow of hot air. 

The cell with a detachable short is cooled by immersing it in a glass with coolant (ice-alcohol mixture), and the S11 measurements are performed while the cell is naturally warmed up to the room temperature, then the coolant is replaced with warm (about 35 °C) water and the measurements are performed with natural cooling down to room temperature. 

The temperature is measured with type K thermocouples soldered to the cells’ external surface close to the measured sample. The measurement accuracy is about ±0.5° for short-circuit-open-circuit measurements at 0 °C; the uncertainty decreases with the increase to the room temperature. The temperature uncertainty for reflection-transmission measurements is about ±2° in the temperature range from −10 to +30 °C and about ±5°at the temperatures below −20 °C.

## 4. Measurement Results and Fitting of the Obtained Data

To shorten the formulae and to simplify calculations, the data treatment is performed in terms of magnetic susceptibility χ=μ−1; the temperature data are presented in Celsius scale.

The measured dependence of complex susceptibility on frequency and temperature for the paraffin-based composite filled with 30% vol of Gd powder (*p_incl_* = 0.3) is presented in Figure 4. 

The composite permittivity does not vary with frequency and temperature within the measurement accuracy and is equal to *ε* ≈ 15. A rough estimation of inclusion shape factor with the Maxwell Garnett mixing model with account for $\varepsilon_{binder}\approx 2.2$ results in $N_{incl}\approx \varepsilon_{binder}\times p_{incl}/\left(1-p_{incl}\right)/\left(\varepsilon -\varepsilon_{binder}\right)\approx 0.07$; the appropriate model can be defined studying the samples with a wide range of filling factors only. 

The dependence of complex permeability on frequency is fitted with Havriliak-Negami formula [28] generalized with account for the resonance term $f/F_{rez}$ Equation (3). The advantage of Equation (3) is that it describes a variety of lineshapes with five fitting parameters only.
(3)$$\chi_{0}(f)=\chi_{0}{\left(1-{\left(\frac{f}{F_{rez}}\right)}^{2\delta }+i{\left(\frac{f}{F_{rel}}\right)}^{\delta }\right)}^{-\alpha }=\chi_{0}{\left(1-{\left(\frac{f}{F}\right)}^{2\delta }+i{\left(\Gamma \frac{f}{F}\right)}^{\delta }\right)}^{-\alpha }$$

Here parameters *α* and *δ* characterize spectrum width and symmetry correspondingly (the parameter limitations are: α>0, δ>0, α×δ≤2), *F_rez_* is resonance frequency, *F_rel_* is relaxation frequency and *χ*_0_ is static susceptibility. The relaxation and resonance frequencies are related by the damping factor: $F_{rel}=F_{rez}/\Gamma$. For α=δ=1 and $F_{rel}>>F_{rez}$ Equation (3) describes the Lorentzian dispersion, and for $F_{rel}<<F_{rez}$, Equation (3) describes Debye dispersion. Parameters *α* and *δ* are included into the fitting together with the Lorentzian resonance parameter $F_{rez}$ to account at least partially for distribution of cluster shapes and therefore to increase the approximation accuracy for the measured data (Figure 4). 

The shape distribution may be described accurately with a Bergman-Milton geometrical spectral function [10], but its determination is a separate task falling out of the research scope, while the accuracy of fitting with Equation (3) is enough to estimate the effect of temperature on reflection and transmission spectra of a microwave screen. 

The measured dependence of static susceptibility on temperature is shown with circles in Figure 5. The real part of microwave susceptibility (Figure 4a) extrapolated to zero frequency is shown with crosses. The difference may be attributed to the higher measurement error with electrically thin samples compared to that of short-circuit measurements with much thicker samples.

The dependence of static susceptibility on temperature is fitted (the grey line in Figure 5) with a Cauchy cumulative distribution function (CDF, Equation (4)); fitting with other distributions with two free parameters result in much higher inconvergence with the measured data.
(4)$$\chi_{eff}\left(\tau \right)=\mu \left(\tau \right)-1=w\times CDF\left[\left(-\tau_{mean},\sigma \right),-\tau \right]=\frac{w}{2}+\frac{w}{\pi }\arctan\left[\frac{\tau -\tau_{mean}}{\sigma }\right]$$

Here *σ* is the standard deviation related to the width of phase transition zone, *τ* is the distribution argument functionally related to the measurement temperature *t*, but t≠τ; $w=\chi_{0{}^{\circ}C}$ is the scaling factor equal in case under study to susceptibility measured at t≈0 °C. 

An extrapolation with Maxwell Garnett formula of the composite ($p_{incl}=0.3$) susceptibility onto susceptibility of the gadolinium inclusion $\chi_{Gd}=\chi_{composite}/\left[p-N_{incl}\chi_{composite}\left(1-p\right)\right]$ with account for the inclusion shape-factor determined from permittivity measurements ($N_{incl}\approx 0.07$) results in the dotted curve in Figure 1. The extrapolation is performed with a simplest mixing model unverified for the composite under study, the model does not account for effects related to structure changes in the composite with increase of filling factor (percolation, inversion of matrix structure, etc. [10]) and for the possible difference in magnetization of bulk metal and that of micrometer size grinded particles. Nevertheless, the estimated susceptibility appears to be close to the data retrieved from the published magnetization curves (Figure 1).

## 5. Cluster Magnetization Model

The fitting of the measured susceptibility with a continuous distribution function is selected in accordance with a cluster magnetization model valid in the vicinity of the Curie temperature [29,30]. The model assumes that magnetic clusters, i.e., assemblies of a multitude of interacting spin centres which are magnetically decoupled from their environment, exist even in a bulk magnet. The number, shape, and intrinsic magnetization of clusters depend on bias and temperature of the magnet relative to its Curie temperature. In developing the cluster model in terms of mixing models, we have to relate the argument of Cauchy distribution −∞<τ<+∞ (Equation (4)) to the measurement temperature 0<t<∞ or the volume fraction of magnetized clusters 0<p<1, to the shape factor 0<N<1 and to intrinsic susceptibility 0<χ<∞ of the clusters. 

In case of a composite, the effective volume fraction of permeable clusters depends on temperature *p*(*t*) and may be equal to the volume fraction of metal inclusions *p_incl_* at temperatures much lower than Curie temperature only. The cluster shape and demagnetizing factor differ from that of metal inclusion $N_{cl}\neq N_{incl}$ as close to Curie temperature according to the cluster model the inclusion is magnetized partially (the clusters are magnetically decoupled from an impermeable metal matrix). 

In order to relate the distribution argument *τ* to the measurement temperature *t* or to the effective volume fraction *p*, it is necessary to renormalize the function argument and to rewrite the distribution function (Equation (4)) correspondingly: (5)$$\chi_{eff}\left(t\right)=\frac{w}{2}+\frac{w}{\pi }\arctan\left[-\frac{1}{\sigma }Log\left[\frac{t+t_{0}}{t_{C}+t_{0}}\right]\right]$$

Here *σ* is the standard deviation, *t* is the measurement temperature, $t_{0}=273^{\circ }$; the distribution mean is reached at the Curie point $t=t_{C}$. In the composite under study $t_{C}=15.5\;{}^{\circ}C$ (see Figure 5), the value is about 5° lower than the Curie temperature reported for bulk gadolinium [21,23].

The effective volume fraction of clusters *p* and the measurement temperature *t* are interchangeable parameters of the distribution function argument *τ* in Equation (4). The argument *τ* is the same for both *p* and *t*, τ=0 corresponds to p=0.5 or $t=t_{C}$; *τ*, *p* and *t* parameters are related: (6)$$\tau =Log\left[\frac{p}{1-p}\right]=-Log\left[\frac{t+t_{0}}{t_{C}+t_{0}}\right]$$

Therefore Equation (4) may be written as well in terms of volume fraction common for mixing models:(7)$$\chi \left(p\right)=\frac{w}{2}+\frac{w}{\pi }\arctan\left[\frac{1}{\sigma }Log\left[\frac{p}{1-p}\right]\right]$$

Here *σ* is the same standard deviation as in Equation (4), parameter *p* is the effective volume fraction of permeable clusters and the maximum distribution density is reached for p=0.5. 

Note that the relation (Equation (6)) between the argument *τ* of Cauchy distribution (Equation (4)) and temperature *t* (Equation (5)) or effective volume fraction of magnetized clusters *p* (Equation (7)) is not the only one; e.g., it is possible to cut off the distribution “tails” which are far from *t_C_* and to reduce the cumulative distribution so that the change of $\chi_{eff}$ will occupy the whole 5 °C<t<25 °C range or the whole filling range 0<p<1. The term “effective volume fraction” is used here to emphasize that *p* depends on the selected functional relation between the temperature *t* and the argument *τ* of the distribution function (Equation (4)). It is a separate task to determine the real volume fraction of magnetized clusters in metal; the effective volume fraction *p* is introduced here only to relate the cluster magnetization model to mixing models and to illustrate the transition from *p* to temperature *t* (Equations (8) and (9)).

Summing all magnetic moments in the sample it is possible to calculate [9] its effective susceptibility as:(8)$$\chi_{eff}\left(p_{1}\right)=\int_{0}^{p_{1}}\frac{\chi \left(p\right)\partial p}{1+N_{cl}\left(p\right)\chi \left(p\right)}=\frac{p\times \chi \left(p\right)}{1+N_{cl}\left(p\right)\chi \left(p\right)}$$

The integral part of Equation (8) is as a matter of fact the Wiener mixing model with an accounting for inclusion shape and the infinite number of mixture components [10]. The integral equation may be solved analytically (the right side of Equation (8)) applying the integral mean theorems. The values χ(p) and $N_{cl}\left(p\right)$ are here the mean values of intrinsic susceptibility and shape factor for permeable clusters in range of cluster volume fractions $0<p<p_{1}$.

Equation (8) is readily written with an account for Equation (6) in terms of measurement temperature *t* instead of the effective volume fraction *p*:(9)$$\chi_{eff}\left(t_{1}\right)=\frac{\chi \left(t\right)}{\left[1+N_{cl}\left(t\right)\chi \left(t\right)\right]}\times \frac{t_{0}+t_{C}}{2t_{0}+t+t_{C}}$$

Similarly to Equation (8) the χ(t) and $N_{cl}\left(t\right)$ are here the mean values of intrinsic susceptibility and shape factor for magnetized clusters in temperature range $t_{0}<t<t_{1}$, $\chi_{eff}$ is the sample susceptibility measured at $t=t_{1}$ or $p=p_{1}$, $t_{C}$ is Curie temperature

Assuming that the magnetic spectra (Figure 4) are symmetric (α≈1) it is possible to simplify Equation (3) and to write it with account of Equation (9) in terms of static susceptibility as function of temperature: (10)$$\chi_{eff}(f,t)=\frac{\chi_{eff}(0,t)}{1-{\left(\frac{f}{F{\left(1+N_{cl}\left(t\right)\chi_{cl}(0,t)\right)}^{1/2\delta }}\right)}^{2\delta }+i{\left(\frac{\Gamma }{{\left(1+N_{cl}\left(t\right)\chi_{cl}(0,t)\right)}^{1/2\delta }}\frac{f}{F{\left(1+N_{cl}\left(t\right)\chi_{cl}(0,t)\right)}^{1/2\delta }}\right)}^{\delta }}$$

Taking into account that the effective filling factor *p* is inversely proportional to the measurement temperature *t* (Equation (6)), the measured resonance frequency and damping factor of gadolinium particle are related to the intrinsic resonance frequency and damping factor of a magnetic cluster as
$$F_{eff}\left(t\right)=F_{cl}{\left[1+N_{cl}\left(t\right)\chi_{cl}(0,t)\right]}^{-1/2\delta }\;\mathrm{and}\;\Gamma_{eff}\left(t\right)=\Gamma_{cl}\times {\left(1+N_{cl}\left(t\right)\chi_{cl}(0,t)\right)}^{1/2\delta }$$

The fitting of the measured dependence of complex susceptibility on frequency and temperature (the dashed lines in Figure 4) shows that the damping factor $\Gamma_{eff}\approx 4÷5$ is too high to determine the resonance frequency $F_{eff}$ accurately (the spectrum is close to Cole-Cole relaxation and $F_{eff}$ exceeds the upper limit of the measurement frequency band), whereas the frequency of the loss peak (the relaxation frequency $F_{rel}=F_{rez}/\Gamma$) is determined readily (the dotted line in Figure 4), therefore it is possible to write the peak frequency in terms of cluster susceptibility and shape factor
(11)$$F_{rel\_eff}\left(t\right)=\frac{F_{cl}}{\Gamma_{cl}{\left[1+N_{cl}\left(t\right)\chi_{cl}(0,t)\right]}^{1/\delta }}=\frac{F_{rel\_cl}}{{\left[1+N_{cl}\left(t\right)\chi_{cl}(0,t)\right]}^{1/\delta }}$$

Equation (11) shows that the increase of the product $N_{cl}\left(t\right)\times \chi_{cl}(t)$ results in the low-frequency shift of the loss peak compared to the peak frequency of magnetized clusters and in the increase of the damping factor. Note that the shift is the opposite of the one observed in composites: the higher the product $N_{incl}\left(p\right)\times \chi_{incl}(p)$, the higher the peak frequency of a composite compared to the inclusion peak frequency [10]. 

The joint solution of Equations (5), (9) and (11) makes it possible to retrieve the dependence of static intrinsic susceptibility $\chi_{cl}(0,t)$ and shape factor $N_{cl}\left(t\right)$ of a mean cluster on temperature. To obtain the continuous lines the discrete data relating the loss peak frequency and temperature presented in Figure 4 are interpolated polynomially. The results are presented in Figure 5 with solid and dashed black lines, respectively.

Theoretically, these very results could be obtained from the microwave measurements (Equation (10)) only; the problem is in the higher temperature error associated with the reflection-transmission measurements compared to the low-frequency short-circuit measurements and in the lower accuracy for susceptibility measurements of electrically thin samples (compare the data in Figure 4 and Figure 5). 

The dependence of a cluster shape factor on temperature ($0\leq N_{cl}\left(t\right)\leq 0.5$, Figure 5) shows that the clusters are fiber-like and that close to the Curie temperature their orientation becomes unrelated to the magnetizing field because of the temperature oscillations of the crystal lattice. The volume fraction, intrinsic susceptibility and demagnetization factor of clusters decrease rapidly with temperature growth above the Curie point. 

The dependence of cluster susceptibility on temperature reveals that the Hopkinson effect [31] (susceptibility peak close to Curie temperature presented by the dashed line in Figure 5) is unobserved in the direct susceptibility measurements (the solid grey line in Figure 5) because of the formation of variety of magnetic clusters (Equation (8)). The shift of the Hopkinson peak and of the shape-factor maximum from the Curie point as well as the shape-factor value slightly below zero at the ends of the temperature range (Figure 5) may be explained by the inaccuracy of microwave spectrum measurements and by the rough approximation of the spectra with the Cole-Cole model (approximate account for the peak frequency with Equation (11) instead of a more accurate account for the resonance frequency, damping and asymmetry factors with Equation (3)).

Preliminary measurements of susceptibility at temperatures below 0 °C show (Figure 4) that the effects unrelated to clusterisation (the freezing of domain walls, the increase of anisotropy field, the transformation of easy axis magnetic anisotropy to the easy plane one [19,32], etc.) dominate there. The effects lead to a decrease of susceptibility with temperature decreases below ~0 °C. This is the reason that the scaling factor *w* in Equations (4)–(7) is assumed to be equal to the susceptibility measured at t≈0 °C. The above effects fall out of the cluster magnetization model (Equations (4)–(11)); their applicability for a tunable screen looks doubtful, therefore the measurements below 0 °C are not analysed here.

## 6. Performance Estimation for a Tunable Microwave Screen

The performance of a microwave screen depends on the Fresnell equations (Equations (14) and(15)) on constitutive parameters *ε* ≈ 15 and *µ* of the composite, on the screen thickness *d*, frequency *f* and load impedance $z_{load}$. The permeability of the composite $\mu =\chi_{eff}+1$ is calculated as the function of temperature *t* and frequency *f* using Equation (10) with the parameters of magnetic spectrum (Equation (3)) measured at *t* = 0 °C (Figure 4); the temperature dependence of resonance frequency *F* and damping factor *Γ* is calculated (Equation (10)) using the cluster susceptibility $\chi_{cl}(t)$ and shape-factor $N_{cl}\left(t\right)$ determined from the quasistatic measurements (Figure 5). The width and symmetry parameters are practically constant (α≈1, δ≈0.7) within the range $5^{\circ }<t<20^{\circ }$.

The absorber performance R(d,f,t) is calculated for the composite screen on a metal substrate ($z_{load}=0$), and the transmission performance T(d,f,t) is calculated for the twice as thick screen in a free space ($z_{load}=1$): (12)$$R\left(d,f,t\right)=\frac{\sqrt{\mu \left(f,t\right)/\varepsilon }\times \mathrm{Tanh}\left[\frac{2\pi if}{c}d\sqrt{\varepsilon \mu \left(f,t\right)}\right]-1}{\sqrt{\mu \left(f,t\right)/\varepsilon }\times \mathrm{Tanh}\left[\frac{2\pi if}{c}d\sqrt{\varepsilon \mu \left(f,t\right)}\right]+1}$$
(13)$$T\left(d,f,t\right)=\frac{Exp\left[\frac{2\pi if}{c}d\left(1-\sqrt{\varepsilon \mu \left(f,t\right)}\right)\right]\times \left[1-{\left(\frac{\sqrt{\mu \left(f,t\right)/\varepsilon }-1}{\sqrt{\mu \left(f,t\right)/\varepsilon }+1}\right)}^{2}\right]}{1-{\left[Exp\left[-\frac{2\pi if}{c}d\sqrt{\varepsilon \mu \left(f,t\right)}\right]\times \frac{\sqrt{\mu \left(f,t\right)/\varepsilon }-1}{\sqrt{\mu \left(f,t\right)/\varepsilon }+1}\right]}^{2}}$$

*c* in the above equations is the light velocity *c* = 3 × 10^8^ m/s.

The dependence of calculated reflection coefficient (in logarithmic scale) on frequency and screen thickness is presented in Figure 6 as contour maps, the darker the filling, the lower the reflectivity. The maps are calculated for the temperature below and above the Curie temperature (0 °C and 20 °C, respectively). 

The reflectivity control range of the composite under study is close to that of the composites with permeable microwires [4], while the transparency tuneability is lower (Figure 7a). The reason is that the temperature increase causes the simultaneous decrease of both real and imaginary permeability parts. The layer loss Exp[2πifcd×Im[εμ(f,t)]] decreases with temperature increase, but the layer specific impedance Re[μ/ε] decreases as well, thus increasing the reflection coefficient from the layer in a free space and decreasing the total effect on the transmission coefficient. Therefore the transparency variation range due to permeability control is in principle lower than the range due to permittivity control [4].

According to Figure 6, the minimal reflectivity of about −20 dB is achieved at 0 °C for an approximately 4.5 mm-thick layer at the frequency of about 3.5 GHz (the position is marked with the circle in the contour maps); at 20 °C the layer reflectivity is about −2 dB. 

The calculated at 3.5 GHz frequency tuneability is shown in Figure 7 (the left graph) as the dependence of reflection (for 4.5 mm-thick layer) and transmission (for 9 mm-thick layer) coefficients on temperature. The right graph in Figure 7 shows the reflectivity curves of 4.5 mm-thick sample measured in the shorted coaxial cell in the frequency range of 2–7 GHz at 5 °C and 25 °C. 

The calculated dependence of reflectivity on frequency and temperature is in agreement with the results of direct measurements (compare the maps in Figure 6 and the curves in Figure 7).

The composite under study may be applied as well as a tunable shield for electromagnetic interference (EMI) suppression. The shield efficiency is proportional to transmission attenuation [33] (see the grey curve in Figure 7a).

## 7. Conclusions

The mechanochemical technique to synthesize a composite filled with micrometer size gadolinium powder that is stable at room conditions is proposed. 

The estimated dependence of the powder permeability on temperature is close to the data retrieved from the published magnetization curves for bulk gadolinium. The measured Curie temperature of the composite is about 5 °C lower that the reported temperature for the bulk metal; the difference may be related to the length limitation of the magnetized cluster by the particle size. 

The composite quasistatic permeability decreases with temperature increase gradually; the main changes take place in the transition range about ±10° from the Curie point, and this range is much wider than the published theoretical estimation (Equation (2)). The effect is interpreted in terms of a cluster magnetization model (Equations (9)–(11)) developed from the Wiener mixing formula. The model explains the dependence of absorption peak frequency and intensity on temperature (Figure 4), and shows that the Hopkinson effect in Gd is masked by cluster magnetization; the effect reveals itself as the peak of intrinsic permeability (Figure 5) of magnetized clusters in the vicinity of the Curie temperature. 

The treatment of the measured composite susceptibility in the frame of the cluster model shows that the cluster shape factor reaches the maximum in the vicinity of the Curie temperature; the maximal value is close to 0.5 (Figure 5), which corresponds to fiber-shaped clusters oriented perpendicular to the magnetic field; at the high- and low-temperature ends of the Curie transition range the shape factor is close to zero, which corresponds to the clusters oriented parallel to the magnetic field.

The model makes it possible to calculate the dependence of a microwave permeability spectrum on temperature and to estimate the performance of the composite layer as a microwave screen or an EMI suppressor shield with temperature tuned attenuation. The reflectivity measurements of a shorted sample are close to the simulated data (Figure 7). 

## Figures and Tables

**Figure 1 sensors-22-03005-f001:**
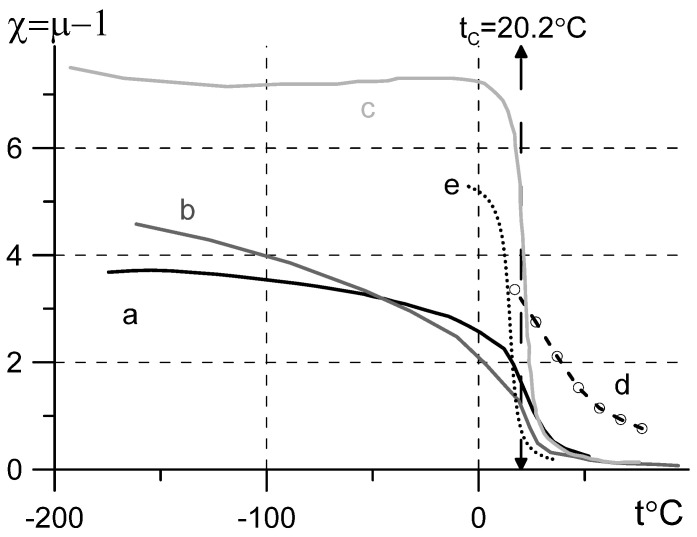
Temperature dependence of Gd susceptibility retrieved from the published magnetization data: curve a is retrieved for bias *H* = 0.4 T as mean value for two orientations of the monocrystalline sample [6]; curve b is retrieved for *H* = 0.5 T [7]; curve c is retrieved for *H* = 0.02 T [8]; curve d is retrieved for *H* = 1.8 T [11]. The dotted curve e shows the susceptibility of the Gd particle retrieved from susceptibility measurements of the composite under study (see Section 3).

**Figure 2 sensors-22-03005-f002:**
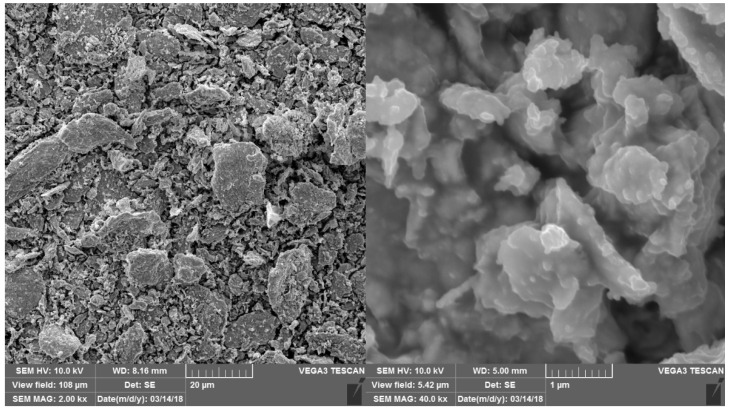
SEM images of Gd powder.

**Figure 3 sensors-22-03005-f003:**
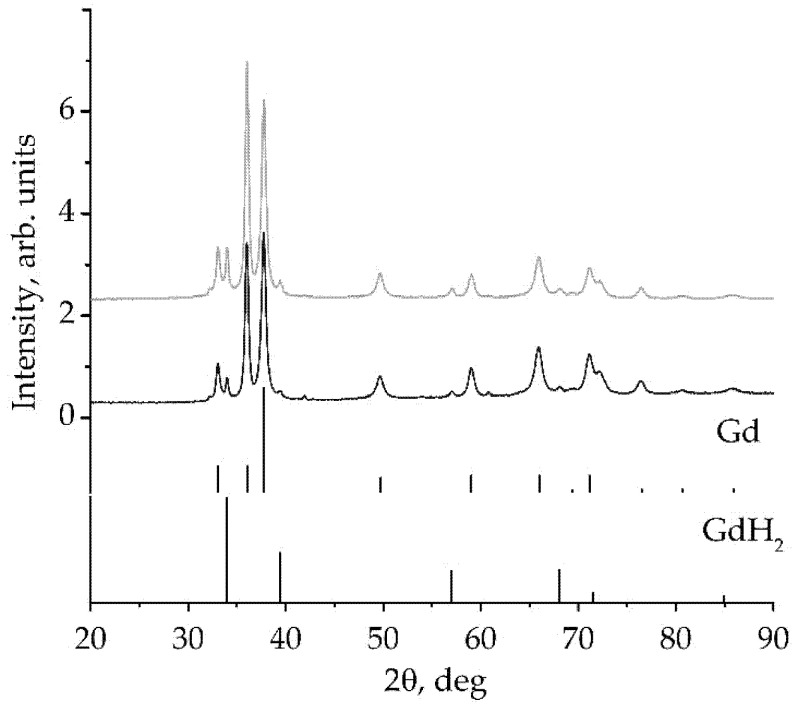
X-ray diffraction powder patterns of Gd powder (using Co Kα radiation) with main peaks indexed. X-axis presents the irraditation angle, Y-axis presents the pattern intensity. The grey top curve corresponds to 2 h grinding, the black bottom one corresponds to 1 h grinding. The lines below the curves mark the tabulated peak positions for Gd and GdH_2,_ respectively.

**Figure 4 sensors-22-03005-f004:**
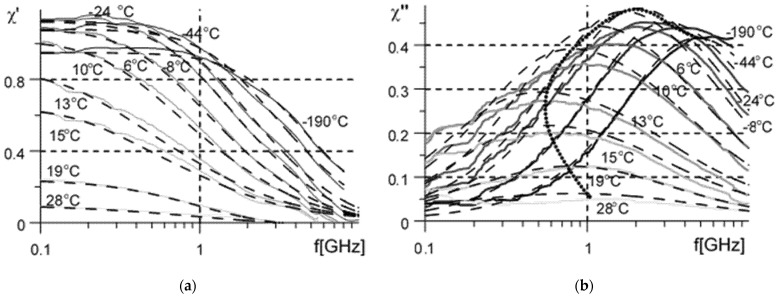
Dependence (**a**) of the real *χ*′ and (**b**) imaginary *χ*″ susceptibility parts on frequency measured at temperatures −190, −44, −24, −8, 0, +6, 10, 13, 15, 19, 28 °C, the higher the temperature, the lighter the line. The thin dashed lines present the approximations with Equation (3). The temperatures close to Curie temperature (10−20 °C) are marked close to the corresponding susceptibility curves, and the dotted line indicates *χ*″ peak position.

**Figure 5 sensors-22-03005-f005:**
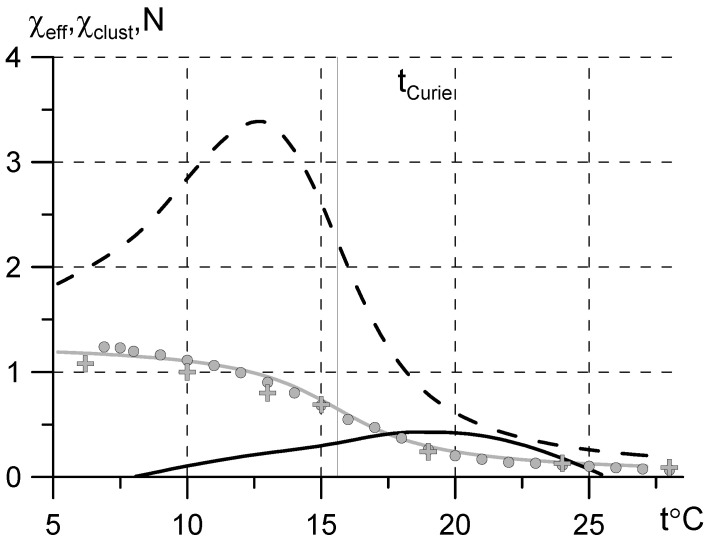
Temperature dependence of the composite static susceptibility (grey line), of cluster shape factor (solid black line), and of intrinsic susceptibility of cluster (dashed black line). The dots and crosses present the measured static susceptibility of the composite and the real part of microwave susceptibility (Figure 4a) extrapolated to zero frequency correspondingly; the lines present simulation data.

**Figure 6 sensors-22-03005-f006:**
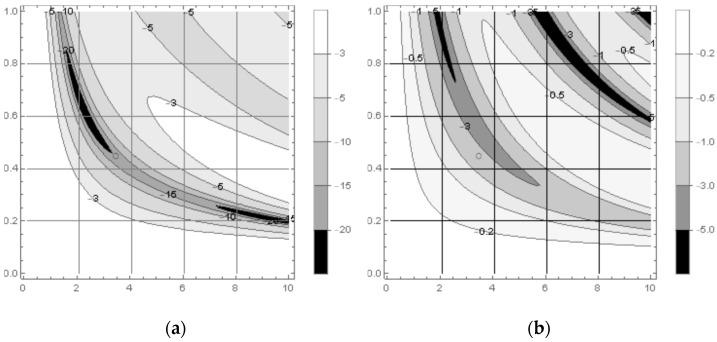
Contour maps for reflection coefficient (dB) of a shorted layer as a function of frequency and layer thickness (*x* and *y* axis correspondingly) calculated for 0 °C (**a**) and 20 °C (**b**).

**Figure 7 sensors-22-03005-f007:**
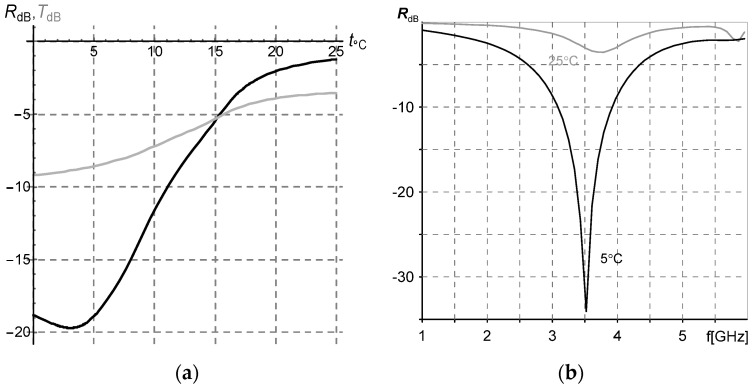
(**a**) Simulated dependence of reflection and transmission coefficients on temperature at f = 3.5 GHz. (**b**) Frequency dependence of reflection coefficient for shorted sample measured at 5 °C and 25 °C.

**Table 1 sensors-22-03005-t001:** The tabulated peak positions, intensities and Miller indexes for Gd and GdH_2_.

1 h			2 h			Gd			GdH_2_		
2θ_exp_	d_exp_	I_exp_	2θ_exp_	d_exp_	I_exp_	d	I	(hkl)	d	I	(hkl)
33.06	3.142	21	33.05	3.144	23	3.145	25	(100)			
34.00	3.059	20	34.00	3.059	14				3.062	100	(111)
36.04	2.891	100	36.00	2.894	94	2.889	25	(002)			
37.76	2.764	84	37.75	2.764	100	2.762	100	(101)			
39.42	2.652	6	39.35	2.656	5				2.652	47	(200)
49.68	2.129	10	49.65	2.130	13	2.127	14	(102)			
57.04	1.873	4	57.05	1.873	3				1.875	30	(220)
59.04	1.815	9	59.00	1.816	17	1.816	16	(110)			
65.94	1.643	17	65.90	1.644	28	1.642	16	(103)			
68.08	1.597	4	68.10	1.597	5				1.599	31	(311)
69.44	1.570	2	69.52	1.569	4	1.572	2	(200)			
71.16	1.537	13	71.15	1.537	24	1.537	16	(112)			

## Data Availability

Not applicable.

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
