# Peer review of "Effect of Temperature on Microwave Permeability of an Air-Stable Composite Filled with Gadolinium Powder"

_sensors, 2022, doi:10.3390/s22083005_

Round 1

Reviewer 1 Report

Referee Report

on paper “Effect of temperature on microwave permeability of an air- stable composite filled with gadolinium powder“ (sensors-1650876) by authors S.N. Starostenko, D.A. Petrov, K.N Rozanov, A.O. Shiryaev and S.F. Lomaeva submitted to Sensors

This is interesting paper. It reports about the features of the magnetic and electrodynamic parameters vs. frequency at different temperatures for Gd-based composites. It was observed giant attenuation (from -2 till -20 dB) at 3.5 GHz in the range 25-5C. This is very important from the fundamental and practical points of view. The obtained experimental results are interesting and reliable. However, paper needs some improvement only after which it can be accepted. At this stage, my decision is minor revision. But I hope that after revision this paper can be accepted. I impressed by the paper.

  1. Abstract:

‑ Please add information (briefly) about the object for investigation – 30 vol. % of Gd powder…and other 70 vol.%?;

‑ I feel that the “shield” can replace “screen” if authors mean protection against EMR (attenuation or shielding).

  1. Introduction:

‑ About eq.(1) – please add information about the range for “N”. How shape impacts on mu? Shape of the sample or shape of the particles? And what is the N value for sphere-rectangle-cylinder shapes?

‑ Page 2 line 54: “Another problem is that monocrystalline gadolinium is anisotropic; the magnetization curve may depend on the sample shape and orientation”. I agree that the orientation of external magnetic field can influences on magnetic characteristics (magnetization, phase transition temperature, coercivity) but please explain me how the shape of the sample correlates with magnetization.

  1. Please highlight the motivation of the concentration range for composite. Why this is such important 30 vol.%.
  2. 3:

‑ Fig. caption “X-ray diffraction patternS”;

‑ There were no any axis titles (Y – intensity; X – 2Q);

‑ Numbers on X-axis are collapsed. Please use larger numbers;

‑ Please add (hkl) for both samples:

‑ Authors provide XRD for Gd-powder after 1 h and 2h grinding, but there is no XRD for investigated composite (30% Gd) – why?

  1. Page 5 line 197 – please add number for eq.
  2. 4:

‑ Please add (a) and (b) on figures and caption for better understanding;

‑ Please add information about the temperatures. Authors use “10”, “13”, “15” and “19” deg. C. But in fig. caption mentioned about the “-190,-44,-8,0,+6,10,13,15,19,28 C” (10 points)

  1. There are some typos and grammatical errors. Please revise this.
  2. My decision is minor revision. But I impressed by the paper. I feel that after brutal revision it can be accepted.

Author Response

We are grateful for a thorough study of the manuscript, for the interest to our research and for the valuable remarks. We have tried to account for all remarks in the revised version of the manuscript; the corrections are yellow-marked in the revised text.

The list of corrections is as follows.

  1.  We have specified in the abstract that the composite under study is prepared of Gd powder and paraffin wax.  We use the terms “screen” because the composite under study appears to be effective as a microwave absorber on a metal substrate with tunable reflectivity, or a window with a tunable transparency, but we have added with a corresponding reference that the Gd-filled composite can be used as tunable EMI shield as well. We are grateful for pointing out an additional application of our study.
  2. We have marked the range for N in the text and have specified that the data in Fig.1 refer to magnetization of the samples of bulk gadolinium. The sample shape affects through the shape factor N the internal field, which is (1-N) times lower than the external field H. This very internal field magnetizes the sample substance, the substance permeability is defined (Eq.1) by the ratio of magnetization M to internal field  H(1-N). The same is valid for a permeable particle in a composite (the relation is the base of mixing models). The sample shape affects the internal field and therefore the measured magnetization effect (see Eq.1 and the above comment)
  3. We have studied the composite with the highest filling factor available, as the low-filled composites are not permeable enough for practice. Obviously the samples with higher filling should be more promising for engineering needs, but it is still problematic technologically to fabricate such composites. We hope to study the samples filled with gadolinium powder up to the percolation threshold in future research.
  4. We have corrected the misprint in Fig.2 caption. We have titled the axes, increased the axes font and added the table with Miller indexes, measured and reference peak position and intensity. The composite under study has been prepared with the powder obtained after 1 hour grinding (section Synthesis of Gd powder, line 8). Binding with paraffin wax affects netiher the powder composition nor the structure; therefore there is no need for additional XRD studies of the composite.
  5. The equation in p.6 is an application of Maxwell Garnett formula for a numerical estimation of the mean shape factor of Gd-inclusions from the permittivity measurements. The equation is unrelated to the developed cluster magnetization model and is not referred to further in the text. We have reformatted the formula to fall in the text line, but left the formula without numeration similar to the formulae in the first line in p.10.
  6. We have marked the left and right graphs in Fig.4,6 as “a” and “b” correspondingly. Only the most essential curves are marked in Fig4 of the original manuscript, now we have marked temperature for all curves in Fig.4

Reviewer 2 Report

Starostenko et al. studied the temperature effects on high-frequency permeability in 30% volume of micrometer-size powder of gadolinium of adaptive composites, and more relative work has been done systematically. The section of conclusion is required to be re-framed and improved, which should contain main findings briefly. As such, the manuscript cannot be published in the present form until some appropriate revisions are corrected.

Author Response

We are grateful for an attentive reading of our manuscript. We have revised the conclusion section: made it more compact and excluded the links to literature from this section

Reviewer 3 Report

This work is devoted to the actual topic. The authors obtained interesting results that may be useful to specialists and therefore can be published.
However, before publication, authors need to carefully edit their manuscript.
1. The order of citing sources is violated (for example, a link to 10 goes earlier than a link to 9, there is no link to 30 source).
2. In line 385, the authors refer to figure 9, but there is no figure 9 in the paper.
3. Links to figures and literature in the Conclusion section leave an unpleasant impression. All this should have remained in the discussion of the results.

Author Response

We are grateful for a thorough study of our manuscript.

The list of corrections is as follows.

  1. We have corrected the publication reference numeration and the erroneous reference to Fig.9 instead of Fig.6.
  2. We have made the conclusion section more compact and excluded links to literature, but we have left the most essential links to equations and figures, as these links simplify the comprehension of research idea just looking through the conclusion section.